# A 34.5 day quasi-periodic oscillation in $\gamma$-ray emission from the blazar PKS 2247–131

Jianeng Zhou[1], Zhongxiang Wang[1], Liang Chen[1,2], Paul J. Wiita[3], Jithesh Vadakkumthani [4], Nidia Morrell[5], Pengfei Zhang[6] & Jujia Zhang[7,8,9]

Since 2016 October, the active galaxy PKS 2247−131 has undergone a $\gamma$-ray outburst, which we studied using data obtained with the Fermi Gamma-ray Space Telescope. The emission arises from a relativistic jet in PKS 2247−131, as an optical spectrum only shows a few weak absorption lines, typical of the BL Lacertae sub-class of the blazar class of active galactic nuclei. Here we report a $\simeq$34.5 day quasi-periodic oscillation (QPO) in the emission after the initial flux peak of the outburst. Compared to one-year time-scale QPOs, previously identified in blazars in Fermi energies, PKS 2247−131 exhibits the first clear case of a relatively short, month-like oscillation. We show that this QPO can be explained in terms of a helical structure in the jet, where the viewing angle to the dominant emission region in the jet undergoes periodic changes. The time scale of the QPO suggests the presence of binary supermassive black holes in PKS 2247−131.

[1] Shanghai Astronomical Observatory, Chinese Academy of Sciences, 80 Nandan Road, 200030 Shanghai, China. [2] University of Chinese Academy of Sciences, 19A Yuquan Road, 100049 Beijing, China. [3] The College of New Jersey, 2000 Pennington Road, Ewing, NJ 08628-0718, USA. [4] Inter-University Centre for Astronomy and Astrophysics, Pune 411 007, India. [5] Las Campanas Observatory, Carnegie Observatories, Casilla 601, La Serena 1700000, Chile. [6] Purple Mountain Observatory, Chinese Academy of Sciences, 8 Yuanhua Road, 210034 Nanjing, China. [7] Yunnan Observatories, Chinese Academy of Sciences, 650216 Kunming, China. [8] Key Laboratory for the Structure and Evolution of Celestial Objects, Chinese Academy of Sciences, 650216 Kunming, China. [9] Center for Astronomical Mega-Science, Chinese Academy of Sciences, 20A Datun Road, Chaoyang District, 100012 Beijing, China. Correspondence and requests for materials should be addressed to Z.W. (email: wangzx@shao.ac.cn)

Galaxies all appear to contain a supermassive black hole (SMBH) at their centers, with a mass usually in the range of ~$10^6$–$10^{10}$ $M_\odot$ (where $M_\odot$ is the solar mass)[1]. The SMBH is sometimes fed by enough matter, accreted through a surrounding disc, to produce strong emissions from the central region of the galaxy that can sometimes outshine the emission from all the stars in the galaxy. These Active Galactic Nuclei (AGN)[2] naturally are objects of great interest, and relativistic jets, launched from the immediate vicinity of the SMBHs, are found to be associated with ~10% of AGN[3]. If a relativistic jet happens to be pointing close to our line of sight, the emission observed from the jet dominates other contributions over nearly the entire electromagnetic spectrum, because of the special relativistic (Doppler) beaming effect. For the same reasons, such AGN, classified as blazars, show rapid and large-amplitude variability[4,5].

Studies of the variability of jets allow us to explore their structures, physical properties, and radiation processes[4,5]. Although rare, one particularly intriguing phenomenon revealed by some of these studies are quasi-periodic oscillations (QPOs). QPOs may be interpreted as an evidence of the binary nature of the central SMBH system, such as in the most well-known case OJ 287[6,7], or reflect the innermost stable orbit of a black hole or oscillation modes of the surrounding accretion disc[8,9]. Therefore, searches for QPOs from blazars have been conducted at different wavelengths. At γ-rays, the Large Area Telescope (LAT) on board the Fermi Gamma-Ray Space Telescope (Fermi) has provided a powerful tool for monitoring blazars at γ-ray energies[10]. Candidate γ-ray QPOs have been discovered;[11–18] these QPOs all have yearly time scales. Here we report our discovery of a possible month-long QPO in γ-ray emission of the blazar PKS 2247−131; the only previous claim of a QPO on this time-scale was reported to have been detected at γ-ray TeV energies during a flare of the blazar Markarian 501 in 1997[19,20], but fewer cycles were observed then and the strength of that oscillatory signal is weak.

## Results

**Quasi-periodic oscillation signal.** In 2016 October, a γ-ray outburst event in the direction of PKS 2247−131 was detected with Fermi LAT[21], and based on multi-wavelength observations, we confirmed the outburst to have originated from PKS 2247−131 (see Methods). Moreover, this source is a blazar-type active galactic nucleus (AGN); more specifically a BL Lacertae sub-type. The γ-ray light curve of PKS 2247−131, constructed by binning 5-day Fermi LAT data in the energy range 0.1–300 GeV, shows a clear periodic modulation after the initial main flux peak (around MJD 57675; see Fig. 1). Our initial search for periodic modulation in the light curve indicated a period of ~34 days, the temporal duration of which ranges from MJD 57693 to 57903 (Fig. 1). These results were obtained from the Weighted Wavelet Z-transform[22] (WWZ) of the entire light curve (see the left panel of Fig. 2). The WWZ method on the light curve during MJD 57693–57903 provided clearer results (see the right panel of Fig. 2). The WWZ power, as well as its time-averaged power, strongly indicates the presence of a ~34 day QPO.

In addition, the Lomb-Scargle periodogram[23,24] (LSP) of the light curve during MJD 57693–57903 provided similar results (see the right panel of Fig. 2). From the LSP power density curve, the oscillation period $P_{obs}$ was determined to be $P_{obs}$=34.5 ± 1.5 days.

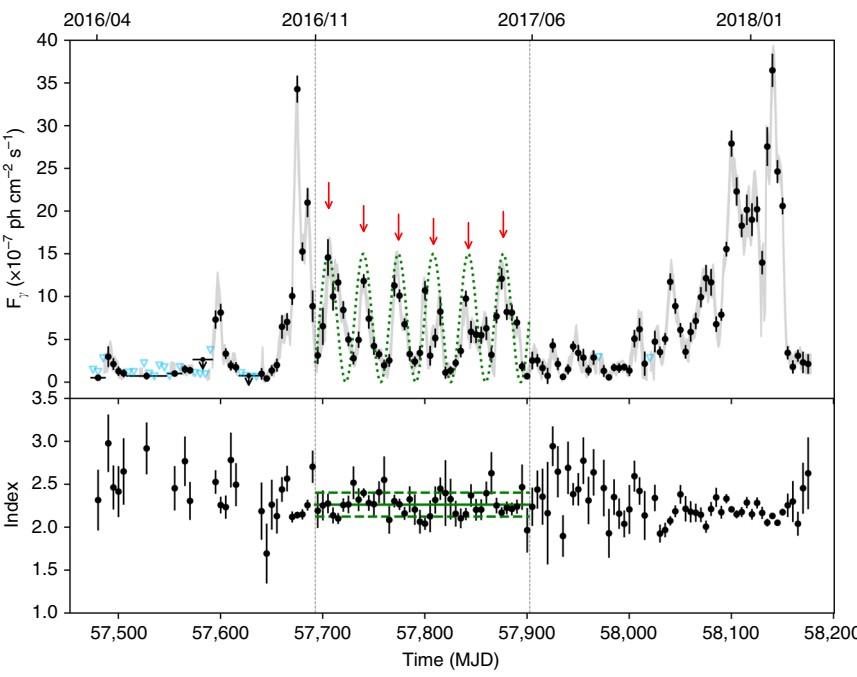

**Fig. 1** Analysis of Fermi LAT data for PKS 2247−131. In the top panel, a 5-day binned light curve of PKS 2247−131 at γ-ray energies of 0.1–300 GeV is shown. Each data point is a flux measurement (with 1σ error bar) obtained from each set of 5-day binned data. Flux upper limits (95% confidence level; light blue downwards triangles) are also shown when the source was not detected in any 5-day binned data (TS<9). For the group of upper limits at the beginning of the light curve, the source can be detected in enlarged time bins (10–40 days; plotted with horizontal bars), while there are still two 20-days upper limits (downwards arrows). Two vertical dashed lines mark the interval MJD 57693–57903 during which clear periodic modulation is seen. For clarity, a smooth light curve, constructed by shifting each 5-day time bin one day forward, is over-plotted as a gray curve. Red arrows and the green dotted sinusoidal curve (with a period of 34.5 days) are plotted to help indicate the modulation in the light curve. In the Bottom panel, we show Photon index values obtained from each set of 5-day binned data, in which a power law is used in the analysis. The green line indicates the average value (2.26) of the photon indices during the temporal interval exhibiting the quasi-periodic modulation, and the green dashed lines mark the standard deviation (0.14) range of the data points

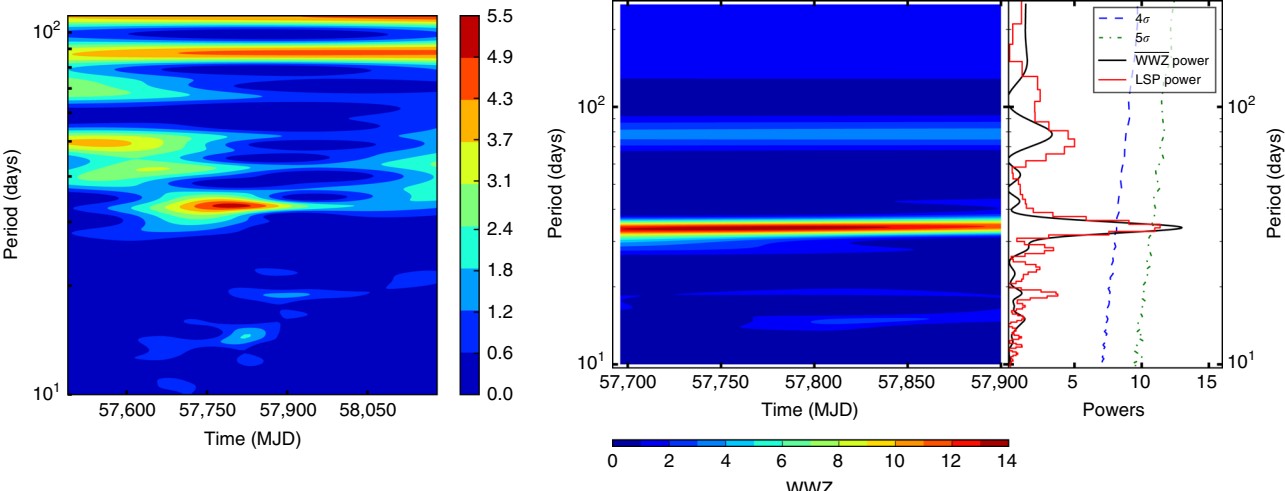

**Fig. 2** Detection of the quasi-periodic oscillation signal. In the left panel, the WWZ power for the whole light curve in Fig. 1 is shown. A periodicity around ~34 days is detected in the time period of MJD 57693–57903, as indicated in Fig. 1. In the right panel, we show the WWZ power and LSP power obtained for the light curve in the time period of the periodic modulation. The black and red curves are the time-averaged WWZ power and LSP power, respectively. The blue dashed and green dash-dotted curves are $4\sigma$ and $5\sigma$ significance curves, respectively, obtained from the light curve simulation. The simulation indicates that the signal has a significance of $5.2\sigma$ (or $4.6\sigma$ considering a trial number of 20)

We used the light curve simulation method[25] to estimate the significance of the oscillation signal, in which the noise in the LSP power density curve was modeled with a smoothly bending power law (see Supplementary Methods and Supplementary Figures 1–3 for details). The $4\sigma$ and $5\sigma$ significance curves were obtained from simulating the light curves $10^7$ times (see the right panel of Fig. 2). The results indicate a significance of $5.2\sigma$ for this signal, or ~$4.6\sigma$ after considering a trial number of 20 (see Supplementary Methods). We thus concluded that a ~$4.6\sigma$ QPO exists in the light curve during MJD 57693–57903. This is the first clear case that a comparatively short, month-long QPO has been seen in $\gamma$-ray emission emanating from a blazar.

Within the time span of the periodic modulation, six full cycles are visible. We modeled the data in each 5-day time bin with a power law $N(E) \sim E^{-\Gamma_{\rm p}}$ (where $E$ and $\Gamma_{\rm p}$ are the photon energy and photon index, respectively), and no significant spectral variations were found. The photon indices during the six cycles have an average of 2.26 with a standard deviation of 0.14, while none of them are significantly away from the $2.26 \pm 0.14$ range (see Fig. 1). We constructed a folded light curve by performing the likelihood analysis on the data of the six cycles in each of 16 phase ranges (phase zero was set at MJD 57692.66; see Fig. 3). The light curve is not symmetric, dropping gradually from the peak to the lowest point of the curve, and a second component appears to exist, as hints of the second component may be seen in the individual modulations. More detailed analysis shows that the second component actually has a flux peak nearly as high as that of the main one in the energy range 1–300 GeV, and the different heights seen between the two components in the folded light curve in the whole energy range are because of somewhat lower fluxes of the second component in the low energy portion, <1 GeV, although their spectral shapes are similar (see Supplementary Figures 4, 5). We note that two Lorentz functions provide a good fit to the light curve profile (Fig. 3; the details of the Lorentz functions are given in Supplementary Table 1).

## Discussion

The recently identified candidate $\gamma$-ray QPOs[11–18], having year-long periods, were often discussed in scenarios involving a binary SMBH AGN system. These are plausible, as if the binary black holes have a total mass of $\sim 10^8\,M_\odot$ and the separation, $R$, between them is at a milli-parsec scale (~100 times the Schwarzschild radius $R_{\rm s} = 2GM/c^2$, where $G$ is the gravitational constant and $c$ is the speed of light, for a mass $M = 10^8\,M_\odot$), the binary orbital period is in the range of several years. The companion black hole could perturb the accretion periodically at the orbital period[7]. Second, the companion black hole could also induce long-term periodicities by exerting a torque on the accretion disc[26]. However, the driven precescion of the accretion disc (and the jet) has a time scale of hundreds of years. On the other hand, single SMBH systems can give rise to short-term, sub-day QPOs, resembling some QPO cases seen in stellar-mass black hole binaries in the Milky Way[8]. Pulsational accretion flow instabilities may also drive jet-disc QPOs, which are predicted to have intra-day time scales[27,28].

The 34.5 day QPO we found in PKS 2247−131 is obviously different from the time scales considered in the above scenarios. A geometrical origin instead is the likely explanation. At radio wavelengths, Very Long Baseline Interferometry (VLBI) has revealed that in some blazars, the parsec-scale cores appear to be misaligned with the large, kiloparsec-scale structures of jets. Such misalignment can be naturally explained by a helical structure in these inner jets[29]. Significant flux variations can arise due to the changing relativistic beaming effect when different parts of such a helical jet pass closest to the line of sight, even if no intrinsic variations of the emission from the jet are present[30]. Furthermore, our viewing angle to the helical motion changes essentially periodically as the emission blob in a jet moves towards us, thus giving rise to QPO-like flux modulations (see a schematic illustration in Fig. 4).

In this geometrical scenario, the viewing angle $\theta$ of an emitting blob's motion as a function of phase $t$ (normalized to the minimum value of $\theta = \theta_{\min}$ at $t = 0$)[31] is $\cos\theta = \sin\phi \sin\psi \cos(2\pi t) + \cos\phi\cos\psi$, where $\phi$ is the pitch angle between the emitting blob's motion and the jet's axis, and $\Psi$ is the inclination angle of the jet to the line of sight (Fig. 4). We then have the relativistic beaming factor $\delta = 1/[\Gamma(1 - \beta\cos\theta)]$, where $\Gamma = 1/\sqrt{1 - \beta^2}$ is the bulk Lorentz factor of the moving blob, which has a bulk speed of $\beta c$. The flux intensity $I(\nu)$ in the observer's frame is transformed from

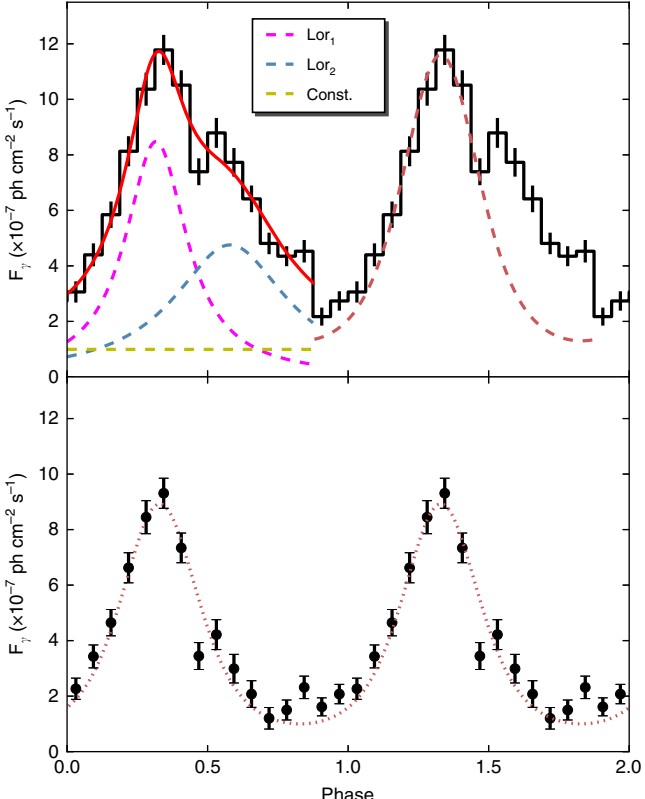

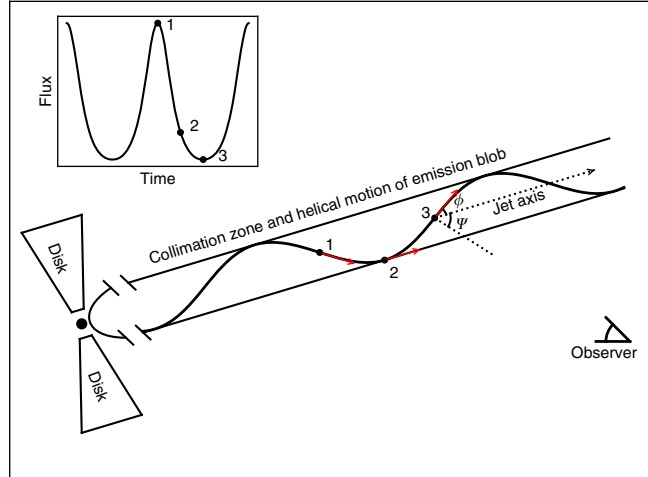

**Fig. 4** Schematic illustration of a helical jet that produces periodically modulated emission. The emitting blob's motion has a pitch angle $\phi$ from the jet's axis, which has an inclination angle $\psi$ from the line of sight. As the emitting blob moves towards the observer, the viewing angle to the blob changes periodically

**Fig. 3** Modulation properties of the quasi-periodic oscillation. In the top panel, the folded light curve, which is constructed from binned likelihood analysis of the six cycles of Fermi data shown in Fig. 1, is plotted as a black histogram, and the error bars are $1\sigma$ uncertainties. Phase zero corresponds to MJD 57692.66 and 16 phase ranges are set. Two cycles are shown for clarity. The red solid curve is a fit to the light curve, which consists of two Lorentz functions (magenta and blue dashed curves) plus a constant (yellow dashed line). The dark-red dashed curve is the modulation curve produced from the simple helical jet model, but it does not fit the right wing of the observed data points. In the bottom panel, we show the main modulation data points when the weak Lorentz function is subtracted. The helical jet model (red dotted curve) can provide a fit to the modulation

$I'(\nu')$ in the blob's comoving frame by[32] $\nu I(\nu) = \delta^4 \nu' I'(\nu')$. We may set $\phi = 2°$, as it usually is in the range of $\leq 1°-4°$[31]. The other parameters that provide a decent fit are $\psi = 5°$ and $\Gamma = 8.5$, both of which are rather typical for blazars. This simple model can produce a light curve that well describes the main part of the observed modulation, but the model light curve is symmetric and does not provide a fit to the right wing of the modulation (Fig. 3). If we consider that there are two components represented by the Lorentz functions used to fit, the model can be adjusted (with only the normalization factor changed) to provide a fit to the observed main modulation (Fig. 3). The origin of the two components is not clear, their difference being the second one has lower fluxes at <1 GeV energies, but we suspect that there are two significant emission regions, which may correspond to a forward shock and a reverse shock[33]. Evidence supporting this geometrical origin over, for example, one involving emission changes resulting from evolution of the emitting particle energies, is provided by the essential constancy of the photon power-law indices (Fig. 1), even when the flux varied periodically by a factor of ~6. We note that this geometrical origin might also be revealed from multi-wavelength observations in optical and X-ray bands; however, the data we have collected are not only too sparse but

unfortunately do not cover the long modulation cycles (see Supplementary Table 2).

Due to the effect of special relativity, the observed period $P_{obs}$ is much shorter than the physical period $P$ at the host galaxy:[34] $P_{obs} = (1-\beta \cos\phi \cos\psi)P$. Using the above parameters, $P \approx 7.1$ years. The distance that the blob moves in one cycle would be approximately $D = c\beta P \cos\phi \approx 2.2$ pc, and the total projected distance in six cycles would be $D_p = 6D\sin\psi \approx 1.1$ pc. Such helical motions of newly born radio emitting knots, along with the parsec-scale jets, have been detected in several blazars[35–37]. For PKS 2247−131, because its distance is large, ~1100 Megaparsec (estimated from the redshift $z \simeq 0.22$; see Methods), the helical motion would not be resolved by VLBI.

A helical structure could be intrinsic in a jet, driven by a coiled magnetic field[38,39]. This structure is supported by optical polarization changes seen in jets[40]. In our case, we note that the putative physical period $P$ is actually on the order of orbital periods of close binary SMBHs, but boosted to appear much shorter, and thus the orbital motion of such a binary could be the cause of the helical motion seen in PKS 2247−131[41]. Due to the orbital velocity $v$ of the jet-launching from the vicinity of the primary black hole, the direction of the ejected blob (at a velocity close to $c$) would have a small angle $\Delta\alpha$ from the axis of the accretion disc (or the black hole spin)[31] $\Delta\alpha \simeq v/c \simeq 1.3° q/(q+1)^{1/2}(R/1000R_s)^{-1/2}$, where $q$ is the mass ratio of the secondary black hole relative to the primary. As a result, the jet appears to be rotating with respect to an observer, and $\Delta\alpha$ corresponds to the pitch angle $\phi$ in the above helical scenario.

Our analysis of the light curve of PKS 2247−131 may appear to be the first time that a jet has been found to exhibit a helical structure through clear flux modulation of its high-energy flux. However, the complexity of the QPOs in blazars or AGN in general must be noted. Our explanation is based only on the monthly modulation time scale, but other QPO driving mechanisms[11,14], such as those mentioned above, may not be absolutely excluded. The binary SMBH possibility is often considered as the cause for yearly QPOs claimed to be found in different searches, particularly at optical bands, but the number of these binaries should be limited, and are constrained by the gravitational wave background[42]. Even in the binary SMBH scenario, the cause of flux modulation in a blazar may be the jet

instabilities, dynamically perturbed by the orbiting companion[43,44].

The helical structure explanation could be confirmed by near-future multi-wavelength monitoring of PKS 2247−131 since the QPO should appear again, although the modulation is extremely sensitive to our viewing angle to the helical motion. Searches for similar QPOs in the γ-ray band can also help as their results could constrain the γ frequency of the appearance of such QPOs. Previous systematic searches in Fermi LAT data have found several candidates with yearly QPOs[45]. Thus far, more than 1500 blazars have been detected with Fermi LAT[10]. We have analyzed most of the LAT data for these known blazars and only 19 of them (see Supplementary Table 3) had γ-ray fluxes as high as that seen during the outburst of PKS 2247−131. However, none of the 19 sources have been found to exhibit similar QPOs. These results may suggest that either PKS 2247−131 is a rather special case because of its putative binary SMBH nature, or similar QPOs are hardly detectable with current high-energy facilities.

## Methods

**Multi-wavelength identification**. PKS 2247−131 has an accurate sky position of RA = $22^h49^m59^s.6125$, Dec. = −12°51′16″.825 (equinox J2000.0), and has been detected by various radio surveys[46] and occasionally optical and infrared surveys, for example the SuperCOSMOS Science Archive[47] and WISE all-sky data[48]. Due to its radio-emission properties and the possibility of being a counterpart to the EGRET γ-ray source 3EG J2251−1341[49], PKS 2247−131 has been considered to be an AGN.

For the outburst starting from 2016 October, 16 Swift X-ray observations of the field containing PKS 2247−131 were conducted since 31 July 2016 (see Supplementary Table 2). We analysed all the Swift XRT data by using its standard data reduction tool xrtpipeline (version 0.13.4). An X-ray source was detected in 11 of the 16 observations. We used one observation (conducted on 2016 October 24) in which the source had the highest count rate, and determined its position to be RA = $22^h49^m59^s.8$, Dec. = −12°51′16″.6 (equinox J2000.0, with a positional uncertainty of 5″.2 at a 90% confidence level). This X-ray position matches well with those of the radio and optical counterparts (see Supplementary Figure 6).

Using the 6.5-meter Clay Telescope, we took two spectroscopic exposures of PKS 2247−131 on 2017 November 22. The exposure times were 20 min and 15 min. The instrument used was the Low Dispersion Survey Spectrograph 3 (LDSS3), with a wavelength coverage of 4250–9500 Å. The spectrum combined from the two exposures is basically featureless, having no emission lines but a few weak absorption lines (∼3–10σ detections; see Supplementary Figure 7). The source thus is a blazar because emission from a jet, arising from non-thermal radiation processes, dominates the host galaxy emission and only weak lines from the host galaxy may be present[50]. We determined the redshift z ≈ 0.22, which implies a source distance of ~1100 Mega-parsec, where the cosmological parameters from the Planck mission[51] are used (the Hubble constant $H_0 \simeq 67$ km s$^{-1}$ Mpc$^{-1}$).

**Fermi LAT data analysis**. We used Fermi LAT data between MJD 54682 (4 August 2008) and MJD 58177 (28 February 2018) in the energy range of 0.1–300 GeV. Events with zenith angles ≤90° were selected, in order to avoid contamination from the Earth's limb. The region of interest (ROI) we set was 20° × 20°, centered at the position of PKS 2247−131. The standard binned likelihood analysis was performed on the data in the ROI. Sources in the Fermi LAT Third Source Catalog[52] and the Galactic and isotropic background (files `gll_iem_v06.fits` and `iso_P8R2_SOURCE_V6_v06.txt`, respectively) were included in our source model.

PKS 2247−131 was not detected by Fermi LAT before April 2016, resulting in a flux upper limit (at a 95% confidence level) of 3.65×10$^{-9}$ ph cm$^{-2}$ s$^{-1}$. Starting from 12 April 2016, the source could be detected in most sets of 5-day binned data. We thus constructed its light curve binned in 5-day intervals, in which the spectral parameters of the sources in the ROI were fixed at their best-fit values, except that the normalizations of the two background components and one variable source within 5° (`Variability_Index>72.44`)[52] were set as free parameters. Detailed flux variations were clearly revealed in the light curve. This choice of 5 day binning provided the shortest intervals for which nearly all bins were long enough to yield detections during the flare. In a few binned data that did not detect PKS 2247−131 (the Test Statistic (TS)[53] values are smaller than 9), we calculated the 95% confidence flux upper limits. We also tried increasing the time bins to 10–40 days for the group of the upper-limit data points before the outburst peak (Fig. 1), but in two 20-day intervals, only upper limits were obtained. In addition, we also constructed a smooth light curve by shifting each 5-day time bin only one day forward (instead of 5 days) and obtaining the flux in such time bins, which helps show fine details of the flux variations. This light curve is over-plotted in Fig. 1.

## Data availability

The data that support the findings of this study are available from the corresponding author upon reasonable request.

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

## Acknowledgements

We thank M. Phillips for sharing the Magellan time, Z.Y. Zheng for helping with the identification of absorption lines in the optical spectrum, and Y. Liu for helping with the timing analysis. This research made use of the High Performance Computing Resource in the Core Facility for Advanced Research Computing at Shanghai Astronomical Observatory. Funding for the Lijiang 2.4 m telescope has been provided by the Chinese Academy of Sciences (CAS) and the People's Government of Yunnan Province. The Lijiang 2.4 m telescope is jointly operated and administrated by Yunnan Observatories and Center for Astronomical Mega-Science, CAS. This research was supported by the National Program on Key Research and Development Project (Grant No. 2016YFA0400804) and the National Natural Science Foundation of China (11603059, 11633007). L.C. acknowledges the support by the CAS grant (QYZDJ-SSW-SYS023). P.Z. acknowledges the support by the National Natural Science Foundation of China (U1738124) and China Postdoctoral Science Foundation (No. 2017M621859).

## Author contributions

J.Z. led the analysis of Fermi data. Z.W. organized multi-wavelength observations and wrote most of the text. L.C. provided theoretical explanations. P.J.W. assisted with theoretical explanations and contributed to the text. J.V. analysed the archival Swift data, and N.M. obtained the optical spectrum data and analysed them. P.Z. helped with the Fermi data analysis, and J.Z. obtained optical imaging data.

## Additional information

**Competing interests:** The authors declare no competing interests.

