## [Peer Review File · Nature Communications]

Reviewers' comments:

Reviewer #1 (Remarks to the Author):

The authors claim that they discovered a periodicity of about one month in the gamma-ray light curve of the blazar PKS 2247-131 and that this suggests emission from a helical jet from a binary black hole system.

I would probably accept (with revision) the paper in another peer-review journal, but if "Nature Communications is committed to publishing important advances of significance to specialists within each field", I honestly cannot recommend this paper for publication in this journal. In the following I explain why.

Major issues

The time series analysis performed to identify the periodicity is based on conventional methods, i.e. Weighted Wavelet Z-transform and Lomb-Scargle periodogram, and nothing new is added.

The theoretical interpretation is just the application of the Sobacchi et al. (2017) model to this specific case. Again, there is nothing new here.

I have seen tens of papers making the same kind of time series analysis and drawing similar conclusions.

The authors claim their case is of particular importance because they find a periodicity of about one month in gamma rays. Actually, a periodicity of the order of 20 days was discovered in the TeV gamma-rays and X-rays data of Mkn 501 and interpreted in terms of a helical jet (Hayashida et al. 1998, ApJ, 504, L71; Rieger and Mannheim 2000, A&A 359, 948). So I cannot agree with the authors when they say that "The case of PKS 2247-131 appears to be the first time that a jet has been found to exhibit a helical structure through clear flux modulation of its high-energy flux."

But then why should periodicity in gamma rays be so special? Gamma-ray radiation likely comes from inverse Compton scattering on optical photons and indeed gamma and optical events are found to be correlated most of the times. Periodicities at optical frequencies have often been found, with a variety of periods, including month-scale ones (see e.g. Lainela et al. 1999, ApJ, 521, 561, who also considered an helical jet interpretation). Sometimes these periodicities have not been found on a longer time basis, i.e. if they were real, they seemed to be transient phenomena. And also in the present case the authors identified 6 peaks over a time span of about 200 days in a light curve of about 700 days. So the interesting question is rather: what are the possible reasons for a transient periodic behaviour?

I am personally convinced that variations of the viewing angle have an important role in shaping blazar light curves, as suggested by the development of instabilities in numerical simulations of jets (e.g. Mignone et al. 2010, MNRAS, 402, 7), theoretical models (e.g. Subramanian et al. 2012, MNRAS, 423, 1707), and radio observations of swirling jets (e.g. Britzen et al. 2017, A&A, 602, A29) and as has been strongly supported by the detection of Doppler factor variations in the recent CTA 102 outburst (Raiteri et al. 2017, Nature 552, 374).

But geometrical interpretations of blazar variability and the existence of binary black hole systems have already been proposed in many papers (see e.g. Ostorero et al. 2004, A&A, 419, 913).

More in detail, the authors adopt a choice of parameters for the model that appears not justified for this specific case. They just use reasonable values or values that imply a (in their words) "decent" fit (to what is not specified). This, again, may be acceptable for a normal paper, but the results that come out of this model cannot be considered "important advances of significance to specialists".

There also seems to be some confusion on the emission mechanism: they apply the Sobacchi et al. model of an emitting blob travelling along a helical jet, but at page 6 they speak of forward and reverse shocks because they need two contributions to reproduce the confused shape of the periodic peaks".

Other issues

The abstract is a bit blatant: the authors made little use of multiwavelength observations; the periodicity analysis was made with gamma-ray data only.

In the first sentence of the Results section they neglect to say that the 2016 gamma-ray flare of this source was announced by the Fermi-LAT Collaboration with ATel 9285 by Buson et al. (2016), who associated it with the radio source PKS 2247-131.

The constancy of the gamma-ray spectral index must be verified on statistical ground.

In the period before the 2016 outburst the authors could try a twice or three times longer time bin to get flux values instead of upper limits.

The helical jet model applied to the data in Figure 3 (bottom panel) does not provide a real good fit: the data seem rather to indicate that the peaks are narrower and that the decomposition into two Lorentz functions in the upper panel must be improved.

Caption of Figure 2: the authors should specify that the absorption lines are produced in the host galaxy.

Finally, the authors say that for clarity they show in Figure 1 a smooth light curve obtained by shifting each 5-day bin forward by one day. I do not understand what they actually did and why.

Reviewer #2 (Remarks to the Author):

General:

I found that the results are interesting, but the amount of work that the authors performed looks rather small, and the absence of confirmation of their result via multi-wavelength observations prevents me from recommending the paper for publication at this time.

The authors analyzed Fermi-LAT observations of one source and discuss evidence for quasi-periodic oscillations observed during a time range of about six months. However, the authors do not explain why this particular source have to show such oscillations. Also, the authors do not explain how they selected the time range of MJD 57693 to 57903. The extension to a longer range will decrease a statistical significance of the evidence for quasi-periodic oscillations. Both these points should be explained.

Major comments about the amount of work and multi-wavelength confirmation are below:

Amount of work:

At the end of the Discussion section, the authors conclude "Since thus far more than 1500 blazars have been detected with Fermi LAT, it is of great interest to check those blazars with sufficiently high gamma-ray fluxes for similar periodicities."

The LAT data is publicly available and the analysis of all the sources from the LAT catalog for a systematic search for quasi-periodic oscillations (including those which are on monthly scales) have already been published, see <https://arxiv.org/abs/1608.06647> and <https://arxiv.org/abs/1707.05829>

Both these papers do not include the data corresponding to a recent gamma-ray outburst from PKS 2247-131 (because they were in press) and also consider the entire data set.

I think that if the authors were to perform a systematic analysis of all the LAT blazars, selecting the bounds of the time interval inside the entire LAT time range (i.e., partial (not entire) data set) and include this analysis in their paper, then the paper will be of higher interest. PKS 2247-131 and PKS 0537-441 (see <https://arxiv.org/abs/1512.08801>) will be two sources showing quasi-periodic behavior in a partial LAT data set.

Multi-wavelength confirmation:

To publish evidence for quasi-periodic oscillation in a high impact journal, the authors should provide evidence through multi-wavelength observations, see e.g.

<https://arxiv.org/abs/1509.02063>

Although the authors performed multi-wavelength identification, there is no multi-wavelength confirmation in the paper under review. To quote the authors, "We note that this geometrical origin might also be revealed from multi-wavelength observations in optical and X-ray bands; however, the data we have collected are not only too sparse but unfortunately do not cover the long modulation cycles".

I propose the authors to perform at least the analysis of LAT data in low- and high-energy bins, e.g. 100-1000 MeV and >1000 MeV, in order to establish quasi-periodic oscillations in both these energy bins.

Red noise:

"the signal significance is 4.7sigma after considering the trial factor in calculating the power curve"

How did the authors approach the problem of the presence of red noise? Are the 4 sigma and 5 sigma significance curves in Figure 2 shown against red noise?

There is no discussion on red noise in the paper under review. It seems that the authors assume a white noise, however the blazar's light curves are characterized by red-noise, see <https://arxiv.org/abs/1004.0348>

Two different approaches are in <https://arxiv.org/abs/1509.02063>

PKS 2247-131: "we identified the outburst to have originated from PKS 2247-131"

I have read the astronomer telegrams about these outbursts and this association, e.g. <http://www.astronomerstelegram.org/?read=9285> and <http://www.astronomerstelegram.org/?read=9620>

Using "we identified" sounds too strong.

Minor points:

Typo: "A geometrical origin instead is the the most likely explanation"

Reviewer #3 (Remarks to the Author):

The paper "A periodicity of 34 days in gamma-ray emission from a blazar" of Jianeng Zhou et al. reports the detection of a half-year-lasting quasi-recurrent oscillations of 34 days in the blazar PKS 2247-131, explained in terms of helical structure miming an orbital binary supermassive black hole motion.

The claim may offer interesting considerations. Findings like these deserve to be published in the Nature Communication. However some concerns raise, which need to be emended before supporting the publication. In detail:

1) As general comment, I kindly recommend to avoid to speak about "periodicities". In sources like blazar that are mostly irregularly variable (as surely the authors know since they have already published works about similar issues on PKS 0301-243 and PKS 0426-380, for example). Why to write "periodicity" if the blazar shows recurrent fluctuations only for a half-year interval (less than 30% of the entire light curve), before they dim slowly in the following months exactly as they were previously absent? One could think to write "transitory periodicity" but these terms contradict each other and makes not sense again. There is rather a quasi-regular oscillatory pattern only during an interval, not apparently associated to a particular state of the source. Periods that are intrinsically transient, unfortunately have not the claim of periodicity, nor the paper title with this word. The title and some sentences may be consequently reworded.

2) The method employed to evaluate this peak significance is doubtful, since it is totally omitted in the paper. The significance of the spectral peaks are assessed relative to a specific model of noise fitting: if this is wrong, then the significance calculations will also be wrong. The applied noise fitting seems inappropriate. The inadequacy of the models is deduced from the shape of significance levels related to the Lomb-Scargle power density spectrum reported in Figure 2, being the best fit noise spectrum not, incomprehensibly, reported. The significance levels do not follow the typical wavelength-structured noise, which blazars are suffering (enhanced low-frequency fluctuations over intervals comparable to the sample length hindering the evaluation of significance), but rather some kind of not suitable "white noise". I tried to reproduce the authors' results from the MJD 57693–57903 light curve using two typical models for fitting the red noise in blazars, a power law model and an autoregressive model 1. Using a procedure taking into account the trial factor (e.g. Guidorzi et al. 2016, A&A, 589, 98, Barret . 2012, ApJ, 746, 131), no significance larger than 95% (80 % in the other case), was obtain. This may be a rather interesting result, but no so meaningful as the claim (5 sigma).

The author have to clearly and carefully described in the paper the procedure used to assess significances against spurious detections.

In addition, it is not clear what the authors mean in relation to the two different values of significance reported.

3) The discussion show an interesting interpretation, in which an orbital binary-black-hole motion could be mimed trough a possible helical motion, like the one supposed to be seen in PKS 2247-131.

However, I suggest to the authors to consider with caution, or simply in a not absolute way, the attribution of oscillations to a binary supermassive black hole scenario. The complexity in interpreting quasi-periodic oscillations in blazars and quasars as due to a binary nature of these sources is discussed e.g. in Sandrinelli et al. 2017, A&A, 600,A132 and arXiv, Sesana et al,2018, ApJ, 856, 42. The authors should think about this and write some additional sentences and open to other possible physical pictures.

For the novelty of the approach (which seems to be still quite raw, as e.g. it does not explain the appearance of fluctuations and their disappearance), the same caution is to be applied when they affirm that the case of PKS 2247-131 appears to be the first case "that a jet has been found to exhibit a helical structure through clear flux modulation of its high-energy flux, and thus has provided a piece of key evidence for this structure using gamma-rays."

4) Concerning the possible interpretations of the quasi-regular fluctuation, the authors state that the discovered "relatively short periodicity most likely indicates the presence of a helical jet structure within this blazar during its outburst event". I think that this work would benefit of a more (partially technical) discussion in comparison of the obtained results within different interpretations and different models (i.e. the already quoted Sobacchi et al. 2017, MNRAS, 465, 161, Tavani et al. 2018, ApJ, 854, 11, just to mention some recent works).

5) The section "Methods" will benefit from a shortening of about 30%.

=====

Minor comments and suggestions

Title:

***See point 1).

***In addition, the name of the source should be introduced in the title, instead of referring the work to a generic blazar

Abstract:

*** It need to be partially but substantially rewritten. Some sentences are unclear and/or written in a not comprehensible way. Some example:

* "The emission arises from a relativistic jet in PKS 2247-131 as an optical spectrum we obtained only shows a few weak absorption lines,..."

The sentence is not clear: could it run in the following way?

The emission arisen from a relativistic jet in PKS 2247-131 as an optical spectrum only shows a few weak absorption lines.

**"Compared to previously identified, greater than year-long quasi-periodicities in blazars, PKS 2247-131 exhibits the first case of a relatively short, month-like periodicity at gamma-ray energies..."

Do the authors mean:

Compared to one-year time-scale quasi-periodicities, previously identified in blazars in Fermi energy ranges, PKS 2247-131 exhibits ... ?

**"...quasi-periodicity can be explained in terms of a helical structure in the jet so that our viewing angle to the dominant emission region in the jet undergoes periodic changes" -

...quasi-periodicity can be explained in terms of a helical structure in the jet, where the viewing angle of the dominant emission region in the jet undergoes to periodic changes. Can it work?

In general, I suggest a overall revision of the text.

Introduction

***The introduction is quite poor, lacking of a complete picture including the latest finding (e.g. Bhatta et al. 2016ApJ, 832,37B; Prokhorov & Moraghan 2017, MNRAS, 471, 3036; Sandrinelli et al. 2017, A&A, 600, A132, and 2018, arXiv 1801.06435) or possible alternative interpretations.

***"QPOs can help indicate the binary nature of.." - QPOs may be interpreted as an evidence of the binary nature

Results:

***" Fitting each spectrum of the data points with a power law" : which spectrum? This sentence deserve to be better explained.

Figure 4

The shape of plot in the sub-box in Figure 4 is supported by the theoretical model? It would be appreciable. Or it is simply the reproduction of the bottom panel in Figure 3?

Only as a suggestion:

The folded light curve presentation in Figure 3 hides useful information, and there is no motivation as to why standard errors are plotted instead of or together the data points (in the background)

>Reviewers' comments:

>Reviewer #1 (Remarks to the Author):

>The authors claim that they discovered a periodicity of about one month in
>the gamma-ray light curve of the blazar PKS 2247-131 and that this
suggests
>emission from a helical jet from a binary black hole system.

>I would probably accept (with revision) the paper in another peer-review
>journal, but if "Nature Communications is committed to publishing
important
>advances of significance to specialists within each field", I honestly
cannot
>recommend this paper for publication in this journal. In the following I
>explain why.

>Major issues

>The time series analysis performed to identify the periodicity is based on
>conventional methods, i.e. Weighted Wavelet Z-transform and Lomb-Scargle
>periodogram, and nothing new is added.

>The theoretical interpretation is just the application of
>the Sobacchi et al. (2017) model to this specific case. Again, there is
>nothing new here.

>I have seen tens of papers making the same kind of time series analysis
>and drawing similar conclusions.

>The authors claim their case is of particular importance because they find
>a periodicity of about one month in gamma rays. Actually, a periodicity of
>the order of 20 days was discovered in the TeV gamma-rays and X-rays data
>of Mkn 501 and interpreted in terms of a helical jet (Hayashida et al.
1998,
>ApJ, 504, L71; Rieger and Mannheim 2000, A&A 359, 948). So I cannot agree
>with the authors when they say that "The case of PKS 2247-131 appears to
be
>the first time that a jet has been found to exhibit a helical structure
>through clear flux modulation of its high-energy flux."

We thank this reviewer for pointing out this early claim.

We have added a mention of this TeV case and references
(Hayashida et al. 1998 and Osone 2006) at the end of Introduction;
however, that claimed QPO has a lower significance than our case.
Also frankly speaking, the TeV light curve seems not very convincing.

The high GeV flux of our source during the outburst and the great
capability
of Fermi/LAT (long-term monitoring and high sensitivity) have allowed an
excellent monthly QPO case to be seen for PKS 2247-131. We stressed that
this is a "clear" case at GeV energies by adding "clear" in the Abstract
(in italics) and a few other places. We believe that this results is
sufficiently new and interesting.

>But then why should periodicity in gamma rays be so special? Gamma-ray
>radiation likely comes from inverse Compton scattering on optical photons
>and indeed gamma and optical events are found to be correlated most of
>the times. Periodicities at optical frequencies have often been found,
>with a variety of periods, including month-scale ones (see e.g.
>Lainela et al. 1999, ApJ, 521, 561, who also considered an helical jet
>interpretation). Sometimes these periodicities have not been found on a
>longer time basis, i.e. if they were real, they seemed to be transient
>phenomena. And also in the present case the authors identified 6 peaks
>over a time span of about 200 days in a light curve of about 700 days.
>So the interesting question is rather: what are the possible reasons for a
>transient periodic behaviour?

>I am personally convinced that variations of the viewing angle have an
>important role in shaping blazar light curves, as suggested by
>the development of instabilities in numerical simulations of jets
>(e.g. Mignone et al. 2010, MNRAS, 402, 7), theoretical models
>(e.g. Subramanian et al. 2012, MNRAS, 423, 1707), and radio observations
>of swirling jets (e.g. Britzen et al. 2017, A&A, 602, A29) and as has been
>strongly supported by the detection of Doppler factor variations in
>the recent CTA 102 outburst (Raiteri et al. 2017, Nature 552, 374).
>But geometrical interpretations of blazar variability and the existence of
>binary black hole systems have already been proposed in many papers
>(see e.g. Ostorero et al. 2004, A&A, 419, 913).

Thanks for sharing these insightful thoughts.

In our view, we have found a clear modulation with a monthly period
at gamma-ray GeV energy, which is the first such case. We do not claim
that

our modeling is original and have cited many papers providing earlier
models

(but cannot cite all due to publication limits).

>More in detail, the authors adopt a choice of parameters for the model
>that appears not justified for this specific case. They just use
reasonable
>values or values that imply a (in their words) "decent" fit (to what is
not
>specified). This, again, may be acceptable for a normal paper, but
>the results that come out of this model cannot be considered "important
>advances of significance to specialists".

>There also seems to be some confusion on the emission mechanism: they
apply
>the Sobacchi et al. model of an emitting blob travelling along a helical
>jet, but at page 6 they speak of forward and reverse shocks because they
>need two contributions to reproduce the confused shape of the periodic
peaks".

Yes. We agree that the emission is not always clean, nor would one expect
it

to be. As shown in Figure 3, the main modulation can be described by
the simple geometrical model and we admit that we can only speculate at
this point about the smaller second structure in the light curve.

We added the folded light curves at the energy ranges of <1 GeV and >1 GeV

and the phase resolved spectra in Supplementary document. These analysis results may be evidence for the existence of two components: during the peak phase of the second component, the fluxes at <1 GeV energy range are slight lower than those during the peak phase of the main component (see below our reply to Reviewer #2).

>Other issues

>The abstract is a bit blatant: the authors made little use of multiwavelength observations; the periodicity analysis was made with gamma-ray data only.

We replaced "multiwavelength observations" with "data obtained with the Fermi..."

>In the first sentence of the Results section they neglect to say that >the 2016 gamma-ray flare of this source was announced by the Fermi-LAT >Collaboration with ATel 9285 by Buson et al. (2016), who associated it with >the radio source PKS 2247-131.

Thanks for pointing this out. We added the reference in the first sentence of the Results Section.

>The constancy of the gamma-ray spectral index must be verified on statistical ground.

We calculated the standard deviation of the indices during the 6 cycles, and added them to Figure 1 (green dashed lines). No indices are significantly away from this range.

We added this information in the sentence there in the 3rd paragraph of Results and the Figure 1 caption (marked in boldface).

>In the period before the 2016 outburst the authors could try a twice or >three times longer time bin to get flux values instead of upper limits.

We tried this, although two 20-days data points are still upper limits. We added these longer time binned data in Figure 1, and mentioned them in the caption. A sentence was also added in the last paragraph of Methods. Both were marked in boldface.

>The helical jet model applied to the data in Figure 3 (bottom panel) does >not provide a real good fit: the data seem rather to indicate that the peaks >are narrower and that the decomposition into two Lorentz functions in >the upper panel must be improved.

Two Lorentz functions were used to fit the folded light curve, and the model curve was calculated, trying to describe the main modulation. We adjusted the model by changing Gamma to 8.5 and the fit looks better. We changed the Gamma value from 7.5 to 8.5 in the 3rd paragraph of Discussion.

>Caption of Figure 2: the authors should specify that the absorption lines are produced in the host galaxy.

We added "from the host galaxy" in Figure 2 caption (in supplementary document).

>Finally, the authors say that for clarity they show in Figure 1 a smooth light curve obtained by shifting each 5-day bin forward by one day. I do not understand what they actually did and why.

This basically is a finer light curve: each data point is derived from 5 day LAT data, but we only shift each data point forward by 1 day (comparing to the normal 5-day light curves that are constructed by shifting 5 days, i.e., no overlapping between two data points). This way, which is sometimes used for cases of having coarse data, can help show more details of a light curve. We overplotted it for the purpose of connecting the 5-day data points and showing the flux variations of the source as well as possible.

We revised the last sentence at the end of Methods to make this clear.

In addition, in order to well describe the white noise (see the added text in the supplementary document), we also used this smooth light curve to construct the power density spectrum.

>Reviewer #2 (Remarks to the Author):

>General:

>I found that the results are interesting, but the amount of work that the authors performed looks rather small, and the absence of confirmation of their result via multi-wavelength observations prevents me from recommending the paper for publication at this time.

>The authors analyzed Fermi-LAT observations of one source and discuss evidence for quasi-periodic oscillations observed during a time range of about six months. However, the authors do not explain why this particular source have to show such oscillations. Also, the authors do not explain how they selected the time range of MJD 57693 to 57903. The extension to a longer range will decrease a statistical significance of the evidence for quasi-periodic oscillations. Both these points should be explained.

The time range for which the QPO is estimated to be present comes from examining the WWZ power (left panel of Figure 2). We revised the first sentence of 2nd paragraph of Results, to make it clear that the Lomb-Scargle periodogram comes from the selected light curve.

>Major comments about the amount of work and multi-wavelength confirmation are below:

>Amount of work:

>At the end of the Discussion section, the authors conclude "Since thus far
>more than 1500 blazars have been detected with Fermi LAT, it is of great
>interest to check those blazars with sufficiently high gamma-ray fluxes
>for similar periodicities."

>The LAT data is publicly available and the analysis of all the sources
>from the LAT catalog for a systematic search for quasi-periodic
oscillations
>(including those which are on monthly scales) have already been published,
>see <https://arxiv.org/abs/1608.06647> and <https://arxiv.org/abs/1707.05829>
>Both these papers do not include the data corresponding to a recent gamma-
ray
>outburst from PKS 2247-131 (because they were in press) and also consider
>the entire data set.

Yes, we were aware of the two papers. We now cited the first one in
the last paragraph of Discussion.

>I think that if the authors were to perform a systematic analysis of all
>the LAT blazars, selecting the bounds of the time interval inside the
entire
>LAT time range (i.e., partial (not entire) data set) and include this
>analysis in their paper, then the paper will be of higher interest.
>PKS 2247-131 and PKS 0537-441 (see <https://arxiv.org/abs/1512.08801>) will
be
>two sources showing quasi-periodic behavior in a partial LAT data set.

The analysis suggested requires large amount of computing time,
particularly
when searching for short period signals. In any case, we have
followed the suggestion and now have analyzed most of the known ~1500
blazars.
Only about 19 of them had fluxes as high as PKS 2247, which implied that
the type of fine light curves (~5-day binned) needed to search for month-
like
period modulations can be done only for that small number. We have
examined
them in the same way we have looked at this source; however, none of them
show promising signals. We have mentioned this additional work in
the last paragraph in Discussion, and this indicates that the QPO case of
PKS 2247 could be either rare or hardly detectable with current high-energy
facilities.

>Multi-wavelength confirmation:

>To publish evidence for quasi-periodic oscillation in a high impact
journal,
>the authors should provide evidence through multi-wavelength observations,
>see e.g. <https://arxiv.org/abs/1509.02063>
>Although the authors performed multi-wavelength identification, there is
no

>multi-wavelength confirmation in the paper under review. To quote the authors,
>"We note that this geometrical origin might also be revealed from
>multi-wavelength observations in optical and X-ray bands; however, the data
>we have collected are not only too sparse but unfortunately do not cover
>the long modulation cycles".

Yes. When we realized that there was an interesting modulation in the light curve, it was too late to organize optical observations (we did obtain a few optical flux measurements, but they were at the end of the modulation).

>I propose the authors to perform at least the analysis of LAT data in
>low- and high-energy bins, e.g. 100-1000 MeV and >1000 MeV, in order to
>establish quasi-periodic oscillations in both these energy bins.

Yes. We did timing analysis to the >1 GeV data, and the signal is still there. However since the significance of the total data is about 5-sigma, there is no strong motivation to try to establish the signal from part of the data.

Instead we constructed the both folded light curves in the two energy bands of <1 GeV and >1GeV, and showed them as a supplementary figure (Figure 1). The modulation is clearly seen in the >1 GeV band, and it can be noted that at >1 GeV, the two modulation peaks (main and second) are approximately equal in height.

In addition, we also obtained the spectra in the two energy ranges. They are shown as the supplementary Figure 2. The purpose of this figure is to show that at >1 GeV, the spectra of the two components are roughly equal, and at <1 GeV, the second one has the same shape as the main one, but with slightly lower fluxes.

We added the results in the last paragraph of Results, and a sentence in the 3rd paragraph of Discussion.

>Red noise:

>"the signal significance is 4.7sigma after considering the trial factor in
>calculating the power curve"

>How did the authors approach the problem of the presence of red noise?
>Are the 4 sigma and 5 sigma significance curves in Figure 2 shown against
>red noise?

>There is no discussion on red noise in the paper under review. It seems
>that the authors assume a white noise, however the blazar's light curves
>are characterized by red-noise, see <https://arxiv.org/abs/1004.0348>

>Two different approaches are in <https://arxiv.org/abs/1509.02063>

Yes. From the experience, since the modulation is clearly seen in the light curve, we believe that the signal is true. Also because of the signal is at a rather low frequency, it is hard to estimate the red noise. In the original manuscript, we did not consider the red noise when using the Lomb-Scargle periodogram (LSP) power peak to estimate the 4.7-sigma significance. In our light curve simulation, the red noise was considered as part of the simulation work, but could have been treated better. In any case, we should have included the analysis in either Methods or Supplementary document.

Now we have re-done the light curve simulation and updated the right panel of Figure 2. The previous fit to the power density spectrum (PDS) was probably not well constrained due to the lack of high frequency signals. We now used the smooth light curve to construct the PDS, which has allowed us to have a more constrained fit. We added a section in the supplementary document about the light curve simulation analysis. We removed the 4.7-sigma sentence, which was directly estimated from the LSP power peak. As pointed out by Reviewer #3 below, two values may be redundantly done. We also revised the text of the second paragraph in Results slightly. The significance is still at 5.2-sigma.

In addition, we provided a figure with this reply, which shows that the LSP signal is at a 99.7% confidence level after considering red noise. The method to produce the figure is similar to the above, but for the red noise (the red curve in the figure), the first order of an auto-regression function was used. We did not include this part in the manuscript but mention it here here to reply to the problem raised (also raised by Reviewer #3).

>PKS 2247-131: "we identified the outburst to have originated from PKS 2247â`131"

>I have read the astronomers telegrams about these outbursts and this >association, e.g. <http://www.astronomerstelegram.org/?read=9285> and ><http://www.astronomerstelegram.org/?read=9620>

>Using "we identified" sounds too strong.

We changed "identified" to "confirmed" and added the first reference, Buson 2016.

>Minor points:

>Typo: "A geometrical origin instead is the the most likely explanation"

Done. Removed one "the".

>Reviewer #3 (Remarks to the Author):

>The paper "A periodicity of 34 days in gamma-ray emission from a blazar"
>of Jianeng Zhou et al. reports the detection of a half-year-lasting
>quasi-recurrent oscillations of 34 days in the blazar PKS 2247-131,
>explained in terms of helical structure miming an orbital binary
>supermassive black hole motion. The claim may offer interesting
>considerations. Findings like these deserve to be published in the Nature
>Communication. However some concerns raise, which need to be emended
before
>supporting the publication. In detail:

>1) As general comment, I kindly recommend to avoid to speak about
>"periodicities". In sources like blazar that are mostly irregularly
variable
>(as surely the authors know since they have already published works about
>similar issues on PKS 0301-243 and PKS 0426-380, for example). Why to
write
>"periodicity" if the blazar shows recurrent fluctuations only for a
>half-year interval (less than 30% of the entire light curve), before they
>dim slowly in the following months exactly as they were previously absent?
>One could think to write "transitory periodicity" but these terms
contradict
>each other and makes not sense again. There is rather a quasi-regular
>oscillatory pattern only during an interval, not apparently associated to
>a particular state of the source. Periods that are intrinsically
transient,
>unfortunately have not the claim of periodicity, nor the paper title with
>this word. The title and some sentences may be consequently reworded.

Yes. We think "quasi-periodic oscillation (QPO)" is a generally
accepted term (e.g., Ackermann et al. 2015 ApJL, 813, L41). As we already
used QPOs in most places (e.g., Introduction), we changed
"periodicity" to QPO in the title, abstract, and other places when
necessary.

We marked the changes in italic when in the Abstract or in boldface when in
the main text.

>2) The method employed to evaluate this peak significance is doubtful,
>since it is totally omitted in the paper. The significance of the spectral
>peaks are assessed relative to a specific model of noise fitting: if this
>is wrong, then the significance calculations will also be wrong. The
applied
>noise fitting seems inappropriate. The inadequacy of the models is deduced
>from the shape of significance levels related to the Lomb-Scargle power
>density spectrum reported in Figure 2, being the best fit noise spectrum
not,
>incomprehensibly, reported. The significance levels do not follow the
typical
>wavelength-structured noise, which blazars are suffering (enhanced
>low-frequency fluctuations over intervals comparable to the sample length
>hindering the evaluation of significance), but rather some kind of not
>suitable "white noise". I tried to reproduce the authors' results from
>the MJD 57693-57903 light curve using two typical models for fitting
>the red noise in blazars, a power law model and an autoregressive model 1.
>Using a procedure taking into account the trial factor

>(e.g.Guidorzi et al. 2016, A&A, 589, 98, Barret . 2012, ApJ, 746, 131),
>no significance larger than 95% (80 % in the other case), was obtain.
>This may be a rather interesting result, but no so meaningful as the claim
>(5 sigma).
>The author have to clearly and carefully described in the paper
>the procedure used to assess significances against spurious detections.
>In addition, it is not clear what the authors mean in relation to the two
>different values of significance reported.

Yes. Thanks for pointing out this lacuna.
See the above reply to the red-noise comment from Reviewer #2.

>3) The discussion show an interesting interpretation, in which an orbital
>binary-black-hole motion could be mimed trough a possible helical motion,
>like the one supposed to be seen in PKS 2247-131.
>However, I suggest to the authors to consider with caution, or simply in a
>not absolute way, the attribution of oscillations to a binary supermassive
>black hole scenario. The complexity in interpreting quasi-periodic
>oscillations in blazars and quasars as due to a binary nature of these
>sources is discussed e.g. in Sandrinelli et al. 2017, A&A, 600,A132 and
>arXiv, Sesana et al,2018, ApJ, 856, 42. The authors should think about
>this and write some additional sentences and open to other possible
physical
>pictures.
>For the novelty of the approach (which seems to be still quite raw, as
e.g.
>it does not explain the appearance of fluctuations and their
disappearance),
>the same caution is to be applied when they affirm that the case of
>PKS 2247-131 appears to be the first case "that a jet has been found to
>exhibit a helical structure through clear flux modulation of its high-
energy
>flux, and thus has provided a piece of key evidence for this structure
using
>gamma-rays."

Yes. We changed the last paragraph in Discussion to two, and added the
points
suggested above.

>4) Concerning the possible interpretations of the quasi-regular
fluctuation,
>the authors state that the discovered "relatively short periodicity most
>likely indicates the presence of a helical jet structure within this
blazar
>during its outburst event". I think that this work would benefit of a more
>(partially technical) discussion in comparison of the obtained results
>within different interpretations and different models (i.e. the already
>quoted Sobacchi et al. 2017, MNRAS, 465, 161, Tavani et al. 2018, ApJ,
854,
>11, just to mention some recent works).

Yes. We changed "most likely" to "possibly" or "likely" in several places
(Discussion).
We removed the sentence (the previous last one) in Introduction.

As replied in 3), we revised the last paragraph. We also added the

references
suggested here.

5) The section "Methods" will benefit from a shortening of about 30%.

We tried revising the text in Methods by cutting non-key information,
although
did not reach 30%.

=====

>Minor comments and suggestions

>Title:

>***See point 1).

>***In addition, the name of the source should be introduced in the title,
instead of referring the work to a generic blazar

We changed periodicity to QPO and added the name of the source.

>Abstract:

>*** It need to be partially but substantially rewritten. Some sentences
are

>unclear and/or written in a not comprehensible way. Some example:

>* "The emission arises from a relativistic jet in PKS 2247-131 as an
optical

>spectrum we obtained only shows a few weak absorption lines,â€¦"

>The sentence is not clear: could it run in the following way?

>The emission arisen from a relativistic jet in PKS 2247-131 as an optical

>spectrum only shows a few weak absorption lines.

Done.

>*"Compared to previously identified, greater than year-long

>quasi-periodicities in blazars, PKS 2247-131 exhibits the first case of a
>relatively short, month-like periodicity at gamma-ray energiesâ€¦"

>Do the authors mean:

>Compared to one-year time-scale quasi-periodicities, previously identified
>in blazars in Fermi energy ranges, PKS 2247-131 exhibits ... ?

Done.

>*"â€¦quasi-periodicity can be explained in terms of a helical structure in
>the jet so that our viewing angle to the dominant emission region in the
jet

>undergoes periodic changes" -

>â€¦quasi-periodicity can be explained in terms of a helical structure in

>the jet, where the viewing angle of the dominant emission region in

>the jet undergoes to periodic changes. Can it work?

Done.

>In general, I suggest a overall revision of the text.

We have made additional changes in various places, particularly in Abstract, Introduction and Discussion.

>Introduction

>***The introduction is quite poor, lacking of a complete picture including
>the latest finding (e.g. Bhatta et al. 2016ApJ, 832,37B; Prokhorov &
Moraghan
>2017, MNRAS, 471, 3036; Sandrinelli et al. 2017, A&A, 600, A132, and 2018,
>arXiv 1801.06435) or possible alternative interpretations.

We added Sandrinelli et al. 2017 and 2018 as references.
Prokhorov & Moraghan was added in the last paragraph of Discussion.
Different theoretical interpretations are mentioned in the first paragraph
of the Discussion.

Basically, variability in blazars is a huge field and even possible QPOs
is a relatively large sub-field. We believe that we have mentioned the
most
crucial papers. In the Introduction we tried pointing out their
importance,
and then focused on gamma-ray QPOs.

>***"QPOs can help indicate the binary nature of.." - QPOs may be
>interpreted as an evidence of the binary nature €|.

Done.

>Results:

>***" Fitting each spectrum of the data points with a power law" : which
>spectrum? This sentence deserve to be better explained.

We changed the sentences slightly, starting with
"We modelled the data in each 5-day time bin with a ..."

>Figure 4

>The shape of plot in the sub-box in Figure 4 is supported by the
theoretical
>model? It would be appreciable. Or it is simply the reproduction of
>the bottom panel in Figure 3?

The shape of the plot is from the model, but the main plot is purely
illustrative (for example, the viewing angle to the jet, etc.).

>Only as a suggestion:

>The folded light curve presentation in Figure 3 hides useful information,
>and there is no motivation as to why standard errors are plotted instead
of
>or together the data points (in the background)

Sorry we did not make this clear in the text and Figure caption (although
the y-title is likelihood flux). The folded light curve was constructed
from

the binned likelihood analysis. Each data point is the flux obtained from a phase range (0.0625) of the LAT data (starting from phase zero), and the error bar is the uncertainty from the analysis.

We added this information in the text (last paragraph of Results) and Figure 3 caption.

Reviewers' comments:

Reviewer #1 (Remarks to the Author):

The authors made a significant effort to revise the manuscript taking into account the points raised by the referees.

I have no further comments.

Reviewer #2 (Remarks to the Author):

The authors have worked on taking the reviewer's comments into account, and the overall quality of the manuscript has improved somewhat. I feel the amount of supporting data still seems quite small, but I understand only so much can be done because, for example, multi-wavelength confirmation can not be obtained now. At least the authors did look at the data in the <1 GeV and >1 GeV Fermi LAT energy bins to make the analysis more complete.

I think the manuscript may be acceptable for publication after one more revision.

Here I provide some minor comments and grammar corrections:

In Supplementary Figure 1, there seems to be quite a large difference in the shape of the folded light curve at energy > 1GeV, but I do not see any comment on how/why it is different in the main text or figure caption. The authors only mention it in the reviewer comments. I think readers of the paper may be interested.

Title:

~34 days > 34.5 day

Introduction:

our discovery of a month-long > our discovery of a possible month-long

Fermi LAT data at the energy range of 0.1-300 GeV > Fermi LAT data in the energy range 0.1-300 GeV

~34 days QPO > ~34 day QPO

(the right panel of Figure 2; see Supplementary text for details) > (See the right hand panel of Figure 2 and Supplementary text for details)

I would like to see a sentence or two explaining why a 5-day binning was used.

Discussion:

periodically around the orbital period > periodically at the orbital period

Mention that the details of the Lorentz functions used are listed in Supplementary Table 1.

jet-launching, primary black hole, the direction > jet-launching from the vicinity of the primary black hole, the direction

as the results of them could constrain > as their results could constrain

Thus far more than 1500 > Thus far, more than 1500

Would it be useful and interesting to readers to list the 19 blazars in a Supplementary table, if

space permits?

Methods:

standard binned likelihood was performed > standard binned likelihood analysis was performed

Figure 1:

time bin by one day forward > time bin one day forward

Figure 2:

which is marked in Figure 1 > as indicated in Figure 1

Light curve simulation:

range of of a > range of a

allowed to extend the high-frequency > allowed for the extension of the high-frequency

Reviewer #3 (Remarks to the Author):

After a careful reading of the new version and the reply, I can not support the publication of the paper in the journal. I clarify my position in the following.

0) As a preliminary consideration, it is not clear to me if the role of the supplementary document is a sort of appendix to be added to the main text, or if it describes some additional investigations dedicated to the referee, but excluded from the paper.

1) The main point is that quantitative tests of the reliability of the claim against the red noise requested by the referees #2 and #3 are not convincing.

1.a) In the paper the authors claim they obtained a PDS 34-days peak 5-sigma significance by using light curve simulation. This, with some differences, is a very ordinary approach (see the great majority of works on the issue quoted by the authors) that try to reproduce the statistical properties of the observed data/theoretical model.

However, again in the revised paper, the authors do not declare (Sect. "Results") against which kind of noise such as high significance is obtained and red noise has never been mentioned in the new version too.

In the supplementary file the reader discovers that the PDS was fitted with a smoothly bending power-law model. This model, with some differences, is one of the commonly used parametric models to fit the frequency-dependent AGN power density spectra (e.g. McHardy et al. 2004, arxiv 1404.0523, arxiv 0311220, arxiv 0910.2706).

On the contrary, in the reply to referees, they wrote that to take into account the red noise a first order of an auto-regression function model was used, yielding a Lomb-Scargle signal at 99.7% (~3-sigma, not 5 as reported in the paper) confidence level for the claimed quasi-periodicity. It is my opinion that there is a sort of confusion on the issue.

1.b) It is not clear if the applied analysis gives the local false alarm probability (single frequency, as I suppose) or it performs the suitable (and properly downscaled) global significance assessment, corrected through the number of independent frequencies scanned in the search for periodicities. As the model of the light curve (a portion with so few points) corresponding to the period (Fig.1) does not fit the data in at least 20% of the 6 periods, honestly the claim of >5-sigma seems too high to represent a global significance. This, also, makes difficult to accept the numerical result without a solid quantitative and well described demonstration.

2) As the detection of the quasi-periodic oscillation is the fulcrum of the work, any other considerations on the paper are, consequently, of minor importance, and are here omitted.

3) As a general comment on the paper, the new version seem globally less innovative than the previous one, probably due to the adaptation to the comments of the referees.

>Reviewers' comments:

>Reviewer #1 (Remarks to the Author):

>The authors made a significant effort to revise the manuscript taking into
>account the points raised by the referees. I have no further comments.

>Reviewer #2 (Remarks to the Author):

>The authors have worked on taking the reviewer's comments into account,
>and the overall quality of the manuscript has improved somewhat. I feel
>the amount of supporting data still seems quite small, but I understand
>only so much can be done because, for example, multi-wavelength
confirmation
>can not be obtained now. At least the authors did look at the data in
>the <1 GeV and >1 GeV Fermi LAT energy bins to make the analysis more
complete.

>I think the manuscript may be acceptable for publication after one more
revision.

>Here I provide some minor comments and grammar corrections:

>In Supplementary Figure 1, there seems to be quite a large difference in
>the shape of the folded light curve at energy > 1GeV, but I do not see any
>comment on how/why it is different in the main text or figure caption.
>The authors only mention it in the reviewer comments. I think readers
>of the paper may be interested.

We had briefly mentioned this at the end of the results section (second to
last
sentence of the previous manuscript), but did not say it clearly.

We revised the sentence there. We also added a sentence in the
Supplementary
Figure 1 caption (in boldface).

>Title:

>~34 days > 34.5 day

Done.

>Introduction:

>our discovery of a month-long > our discovery of a possible month-long

Done.

>Fermi LAT data at the energy range of 0.1-300 GeV > Fermi LAT data in
>the energy range 0.1-300 GeV

Done.

>~34 days QPO > ~34 day QPO

Done.

>(the right panel of Figure 2; see Supplementary text for details) >
>(See the right hand panel of Figure 2 and Supplementary text for details)

We revised the sentence there, which made the suggested changes.

>I would like to see a sentence or two explaining why a 5-day binning was used.

Basically we tried to employ the finest possible time bins while avoiding having too many upper limits in the light curve.

We added a few words and two sentences in the last paragraph of Methods to clarify this.

>Discussion:
>periodically around the orbital period > periodically at the orbital period

Done.

>Mention that the details of the Lorentz functions used are listed in
>Supplementary Table 1.

We added this at the end of Results.

>jet-launching, primary black hole, the direction > jet-launching from
>the vicinity of the primary black hole, the direction

Done.

>as the results of them could constrain > as their results could constrain

Done.

>Thus far more than 1500 > Thus far, more than 1500

Done.

>Would it be useful and interesting to readers to list the 19 blazars in
>a Supplementary table, if space permits?

Done. We added a new Supplementary Table 2 with this information.

>Methods:
>standard binned likelihood was performed > standard binned likelihood
>analysis was performed

Done.

>Figure 1:
>time bin by one day forward > time bin one day forward

Done.

>Figure 2:
>which is marked in Figure 1 > as indicated in Figure 1

Done.

>Light curve simulation:
>range of of a > range of a

Done.

>allowed to extend the high-frequency > allowed for the extension of the
high-frequency

Done.

>Reviewer #3 (Remarks to the Author):

>After a careful reading of the new version and the reply, I can not support
>the publication of the paper in the journal. I clarify my position in
>the following.

>0) As a preliminary consideration, it is not clear to me if the role of
>the supplementary document is a sort of appendix to be added to the main
>text, or if it describes some additional investigations dedicated to
>the referee, but excluded from the paper.

It is intended as a type of appendix. According to the "Brief guide" of
Nature Communications, a Supplementary document can contain figures, tables
and text that serve as supporting materials for the Main manuscript.
The document is under review, and its figures, tables, and text are linked
in the main text when people read the manuscript on-line.

It is for all readers if they are interested in additional,
supporting materials which often contain details of the data analysis
or additional results.

>1) The main point is that quantitative tests of the reliability of
>the claim against the red noise requested by the referees #2 and #3 are
>not convincing.

>1.a) In the paper the authors claim they obtained a PDS 34-days peak
>5-sigma significance by using light curve simulation. This, with some
>differences, is a very ordinary approach (see the great majority of works
>on the issue quoted by the authors) that try to reproduces the statistical
>properties of the observed data/theoretical model.

>However, again in the revised paper, the authors do not declare
>(Sect. "Results") against which kind of noise such as high significance
>is obtained and red noise has never been mentioned in the new version too.
>In the supplementary file the reader discovers that the PDS was fitted
>with a smoothly bending power-law model. This model, with some
differences,

>is one of the commonly used parametric models to fit the frequency-
dependent

> AGN power density spectra (e.g. McHardy et al.2004, arxiv 1404.0523, arxiv
>0311220, arxiv 0910.2706).

>On the contrary, in the reply to referees, they wrote that to take into

>account the red noise a first order of an auto-regression function model
>was used , yielding a Lomb-Scargle signal at 99.7% (~3-sigma, not 5 as
>reported in the paper) confidence level for the claimed quasi-periodicity.
>It is my opinion that there is a sort of confusion on the issue.

Sorry for the confusion. Our previous reply probably was not clear enough,
because we combined our replies to comments from two reviewers.

We used the light curve simulation method of Emmanoulopoulos et al.
to establish the significance of the QPO signal; this is one of
the best approaches we are aware of for quantifying significance.
Details are provided in the supplementary file.

Because of the following comment in the previous review:

>>suitable "white noise". I tried to reproduce the authors' results from
>>the MJD 57693-57903 light curve using two typical models for fitting
>>the red noise in blazars, a power law model and an autoregressive model
1.

We also tried the auto-regression function model (just for the purpose of
replying to the above comment). The result, the figure that contains the
LSP
signal and the red-noise curve from this model, was attached as part of our
first reply.

We did say in our last reply to reviewer #2:

"In addition, we provided a figure with this reply, which shows that
the LSP signal is at a 99.7% confidence level after considering red noise.
The method to produce the figure is similar to the above, but for
the red noise (the red curve in the figure), the first order of an
auto-regression function was used. We did not include this part in
the manuscript but mention it here here to reply to the problem raised
(also raised by Reviewer #3)."

To improve this aspect, we added a clause in the 3rd sentence of the 2nd
paragraph of Results: "in which the noise in the LSP power density curve
was modeled with a smoothly bending power law." Then we referred to
the supplementary text.

>1.b) It is not clear if the applied analysis gives the local false alarm
>probability (single frequency, as I suppose) or it performs the suitable
>(and properly downscaled) global significance assessment, corrected
>through the number of independent frequencies scanned in the search for
>periodicities. As the model of the light curve (a portion with so few
>points) corresponding to the period (Fig.1) does not fit the data in at
>least 20% of the 6 periods, honestly the claim of >5-sigma seems too high
>to represent a global significance. This, also, make difficult to accept
>the numerical result without a solid quantitative and well described
>demonstration.

Thank you for pointing this out. The significances showed were indeed
local.

We have now calculated the after-trial significance for the LSP signal.
The data for the modulation part in the light curve has a time length of
210 days (MJD 57693 to 57903), and were binned into 42 5-day bins. The
number

of independent frequencies (the trial number) up to the Nyquist frequency was thus $(42/2 - 1) = 20$. After considering this trial number, the significance is lowered to 4.6-sigma.

Therefore we added "or $\sim 4.6\sigma$ after considering a trial number of 20." in the 2nd paragraph of Results (also added in Figure 2 caption). We revised the next sentence there accordingly.

We also added text about the trial number information at the end of Supplementary text.

>2) As a the detection of the quasi-periodic oscillation is the fulcrum of >the work, any other considerations on the paper are, consequently, of >minor importance, and are here omitted.

>3) As a general comment on the paper, the new version seem globally less >innovative than the previous one, probably due to the adaptation to >the comments of the referees.

Although the modifications made in response to the comments of the reviewers have led to some changes that one might consider to have reduced the strength of some aspects of this paper, we disagree that it is less globally innovative. This paper reports a month-long QPO in a GeV flare of a blazar. This phenomenon, a clear case thanks to Fermi, is seen for the first time. We provide a possible explanation, the discussion for which has been extended based on the suggestions of the referees. Particularly thanks to the suggestion from reviewer #2, we have established that the month-long QPO is a rare case among more than 1500 blazars. We believe that the key results remain while the improved discussion and addition of supplementary material have strengthened this paper.

REVIEWERS' COMMENTS:

Reviewer #3 (Remarks to the Author):

The author revised the manuscript in the directions indicated by the referees. In my opinion, no conceptual objection prevents publication of the paper. The main result (the high significance attributed to the peak after frequency-dependent noise modeling) appears to my eyes and experience still rather optimistic (see previous reports). In the respectful acceptance and appreciation of the authors' work, I would recommend further careful checks.

>REVIEWERS' COMMENTS:

>Reviewer #3 (Remarks to the Author):

>The author revised the manuscript in the directions indicated by the referees.

>In my opinion, no conceptual objection prevents publication of the paper.

>The main result (the high significance attributed to the peak after frequency-dependent noise modeling) appears to my eyes and experience still

>rather optimistic (see previous reports). In the respectful acceptance and

>appreciation of the authors' work, I would recommend further careful checks.

Yes. We have carefully checked our methods and results.